# High biodegradability of water-soluble organic carbon in soils at the southern margin of the boreal forest

Yuqi Zhu[1,2], Chao Liu[3], Rui Liu[1,2], Hanxi Wang[1,2], Xiangwen Wu[1,2], Zihao Zhang[1,2], Shuying Zang[1,2*], Xiaodong Wu[1,2,4*]

[1]Heilongjiang Provincial Key Laboratory of Geographical Environment Monitoring and Spatial Information Service in Cold Regions, Harbin Normal University, Harbin, 150025, China

[2]Heilongjiang Province Collaborative Innovation Center of Cold Region Ecological Safety, Harbin 150025, China

[3]School of Resources and Environment, Northeast Agricultural University, Harbin 150030, China

[4]Cryosphere Research Station on Qinghai-Tibet Plateau, Key Laboratory of Cryospheric Science and Frozen Soil Engineering, Northwest Institute of Eco-Environment and Resources, Chinese Academy of Sciences, Lanzhou 730020, China

*Correspondence to*: Xiaodong Wu (wuxd@lzb.ac.cn), Shuying Zang (zsy6311@hrbnu.edu.cn)

**Abstract.** Water-soluble organic carbon (WSOC) is an important component of the soil organic carbon pool. While the biodegradability and its compositional changes of WSOC in deep soils in boreal forests remain unknown. Here, based on spectroscopic techniques, we conducted a 28-day laboratory incubation to analyze the molecular composition, biodegradability, and compositional changes of WSOC during a laboratory incubation for deep soils at the southern boreal margin. The results showed that in the upper 2 m soils, the average content of biodegradable WSOC was 0.228 g kg$^{-1}$ with an average proportion of 86.41% in the total WSOC. In the soil layer between 2.0-7.4 m, the average biodegradable WSOC content was 0.144 g kg$^{-1}$, accounting for 80.79% of the total WSOC. Spectroscopic analysis indicates that the WSOC in the upper soils is primarily composed of highly aromatic humic acid-like matter with larger molecular weights than those in deep soils. Both the aromaticity and molecular weight decrease with depth, and the WSOC is mainly composed of fulvic acid-like matter in the deep soils, suggesting high biodegradability of WSOC in the deep soils. Overall, our results suggest that the water-soluble organic carbon in the boreal forests exhibits high biodegradability both in the shallow layer and deep soils.

## 1 Introduction

Boreal forests cover only around 11% of Earth's land surface, while they store one-third of the global terrestrial carbon stock (Adamczyk, 2021), and substantial amounts are also present in deep layers (Bockheim and Hinkel, 2007; Strauss et al., 2017; Schirrmeister et al., 2011). Climate change can

influence carbon release and sequestration in these soils (Ohlson et al., 2009; Liang et al., 2024), for example, through the melting of ground ice, the occurrence of wildfires, and rising soil temperatures (Zhong et al., 2023; Gao et al., 2021; Zhang et al., 2023; Bond-Lamberty et al., 2007; Kasischke et al., 1995). These changes also alter the composition of soil microbial communities, affecting their stability and functional capacity, and ultimately leading to the loss of organic carbon in northern ecosystems (Zhong et al., 2023; Wu et al., 2021).

Water-soluble organic carbon (WSOC) is a complex mixture composed of both high- and low-molecular-weight compounds, derived from vegetation, litter, root exudates, and microbial biomass and enzymes (Thurman, 1985; Guggenberger and Zech, 1994). It serves as an important substrate for microbial activity (Neff and Asner, 2001; Moore, 2003). Biodegradable WSOC (BWSOC) denotes the portion of water-soluble organic carbon that can be utilized and metabolized by microorganisms (Khan et al., 1998; Marschner and Kalbitz, 2003; Scaglia and Adani, 2009; Vonk et al., 2015). The bioavailability of WSOC largely depends on its chemical composition: simple organic compounds such as amino acids, carbohydrates, and fatty acids are more easily decomposed, whereas more complex components like humic substances require longer decomposition times (Ma et al., 2019). Most leachates from litter and vegetation are dominated by low-molecular-weight molecules, which are highly biodegradable and support microbial growth (Michalzik et al., 2003). Although WSOC accounts for only about 1% of soil organic carbon (SOC) (Margesin, 2008), it represents the most mobile and bioavailable fraction of SOC (Kaiser and Kalbitz, 2012). Climate change can enhance the release of soil carbon as dissolved organic carbon (DOC) into surface water (Bowden et al., 2008; Olefeldt and Roulet, 2012). Understanding the dynamics of this carbon fraction is critical for elucidating SOC turnover in boreal forests (Olefeldt et al., 2014; Öquist et al., 2014).

Due to the cold temperatures, the decomposition rate of soil organic matter (SOM) in boreal forests is low due to the low soil microbial activity (Walz et al., 2017). Over millennial timescales, frozen conditions and cryopedogenic processes, such as cryoturbation, have buried organic-rich surface soils into deep layers, further reducing decomposition rates and promoting long-term carbon sequestration (Ping et al., 2015). These low decomposition rates result in a high proportion of labile and biodegradable fractions within soil organic carbon in boreal forests (Song et al., 2020), including water-soluble organic carbon (Cory et al., 2013). Studies indicate that WSOC in shallow boreal forest soils is highly

biodegradable (Panneer Selvam et al., 2016), with its bioavailability ranging from 24% to 71% (Ma et
al., 2019). However, most of the previous studies focused on WSOC in runoff or soil water rather than
in situ conditions, leaving significant knowledge gaps that hinder our ability to predict SOC loss under a
warming climate.
Previous studies in permafrost regions showed that several factors can significantly influence the
concentration, aromaticity, molecular weight, and optical characteristics of dissolved organic matter
(DOM) (Kurashev et al., 2024). For instance, a freeze–thaw manipulation in a continuous permafrost
region of northern Sweden showed that WSOC biodegradability increased as the freezing front deepened,
largely because protein-like compounds accumulated at this depth (Panneer Selvam et al., 2016). The
chemical nature of WEOM can be an important factor affecting the decomposition of SOM (Paré and
Bedard-Haughn, 2013). In addition, hydrological-redox status can jointly control the stability of SOM
(Pengerud et al., 2013). The tabular ground ice contains a high proportion of labile DOC that may
accelerate the decomposition of permafrost SOM during melting (Semenov et al., 2024). These studies
improved our understandings of DOM in permafrost regions, while few studies have been conducted in
the southern boundary area of boreal forest, which may represent the future conditions of vast boreal
forests due to the climate warming.
Many studies have been conducted to reveal the SOM characteristics within 3 m soils. In a 0–3 m
permafrost profile in the Kolyma River Basin in Siberia, it was found that SOM in permafrost contain
more water-soluble substrates and, after thaw, can be rapidly degraded by active microbes (Uhlířová et
al., 2007). In a Northeast Siberia area, the active layer in around 60 cm, and it was found that the SOM
from permafrost within 1 m depth was more sensitive to temperature changes than that of active layer
(Walz et al., 2017). Since soil deep than 3 m in permafrost regions constitute a large proportion of
permafrost carbon pools, and this carbon pool may also contribute to the future soil organic carbon cycle
(Schuur et al., 2022), it is necessary to understand the SOM dynamics deeper than 3 m depth.
Microorganisms play a key role in the carbon cycle and strongly influence the biodegradability of
WSOC (Marschner and Kalbitz, 2003; Kalbitz et al., 2003a; Neff and Asner, 2001; Yano et al., 2000).
Microbial biomass is more abundant in surface horizons, where soil-organic-carbon mineralization
proceeds rapidly (Henneron et al., 2022; Pei et al., 2025), whereas thaw-activated bacteria in deeper
layers can rapidly mineralize WSOC after permafrost thaws (Drake et al., 2015). Microbial use of WSOC

is modulated by environmental factors such as soil moisture (Zhang et al., 2024; Li et al., 2020) and soil physicochemical properties (Lv et al., 2024; Shao et al., 2022a). Therefore, detailed knowledge of the content, chemical composition, and biodegradability of WSOC along deep soil profiles is critical for clarifying how subsurface carbon is mobilized, transformed, and ultimately influences carbon cycling in boreal forests. The objective of this study is to quantify WSOC of soil profile that deeper than 3 m in a southern boreal margin. We conducted laboratory incubation experiments to determine differences in biodegradable water-soluble organic carbon (BWSOC) and employed spectroscopic techniques to reveal its compositional characteristics (Kothawala et al., 2014; Chavez-Vergara et al., 2014; Sun et al., 2022; Murphy et al., 2008; He et al., 2023). The results can improve our understandings of SOC in boreal forests under a warming climate.

**2 Materials and methods**

**2.1 Study area and sample collection**

The southern region of the boreal forest is highly sensitive to climate warming (Randerson et al., 2006; Zou and Yoshino, 2017; Peng et al., 2022). The forests of the Daxing'an Mountains in Northeast China represent the southernmost extent of the boreal forest biome (Jiang et al., 2002; Huang et al., 2010). The sampling site (50°24'10.8"N, 120°50'12.9"E) is located within the island permafrost zone (Bockheim, 2006; Ran et al., 2012; Brown et al., 1997) (Fig. 1). In 2023, the mean average temperature was -1.24°C, and the annual precipitation of 290.3 mm (Qweather, https://www.qweather.com/en/historical/ergun-101081014.html). The dominant tree species in the study area is *Betula platyphylla*, which characterizes the typical local forest ecosystem (Zou and Yoshino, 2017; Jiang et al., 2002).

During July 9th-11th, 2023, a soil column (13.5 cm in diameter) was collected from a piedmont terrace at an elevation of 734 m. The column extended to a depth of 740 cm and was divided into 12 layers (L1–L12). Soil texture was determined in the field using the "texture-by-feel" estimation method (Vos et al., 2016). Soil color was recorded using the Munsell Soil Color Chart (Table 1). There is structural ice in at the depth between 160-180 cm. Although we could not verify whether this area has permafrost because we lack the ground monitoring data, this site represents the southern margin of the boreal forests.

**Table 1. Depths, soil colors (Munsell color system), textures (based on "texture-by-feel" estimation) of soil samples**

| Named | Depth | Soil color | Soil texture |
|---|---|---|---|
| L1 | 0-10 cm | 10YR 2/1 | Heavy loam |

| | | | |
|---|---|---|---|
| L2 | 10-20 cm | 10YR 2/1 | Heavy loam |
| L3 | 20-30 cm | 10YR 2/1 | Heavy loam |
| L4 | 30-60 cm | 7.5YR 2.5/1 | Silty clay Loam |
| L5 | 60-90 cm | 7.5YR 2.5/1 | Silty clay Loam |
| L6 | 90-120 cm | 7.5YR 2.5/1 | Silty clay Loam |
| L7 | 120-150 cm | 7.5YR 2.5/1 | Silty clay Loam |
| L8 | 150-160 cm | 7.5YR 2.5/1 | Silty clay Loam |
| L9 | 160-180 cm | 7.5YR 2.5/1 | Silty clay Loam |
| L10 | 220-250 cm | 7.5YR 5/2 | Silty clay Loam |
| L11 | 420-450 cm | 10YR 4/3 | Sandy clay |
| L12 | 700-740 cm | 10YR 4/3 | Sandy clay |

117

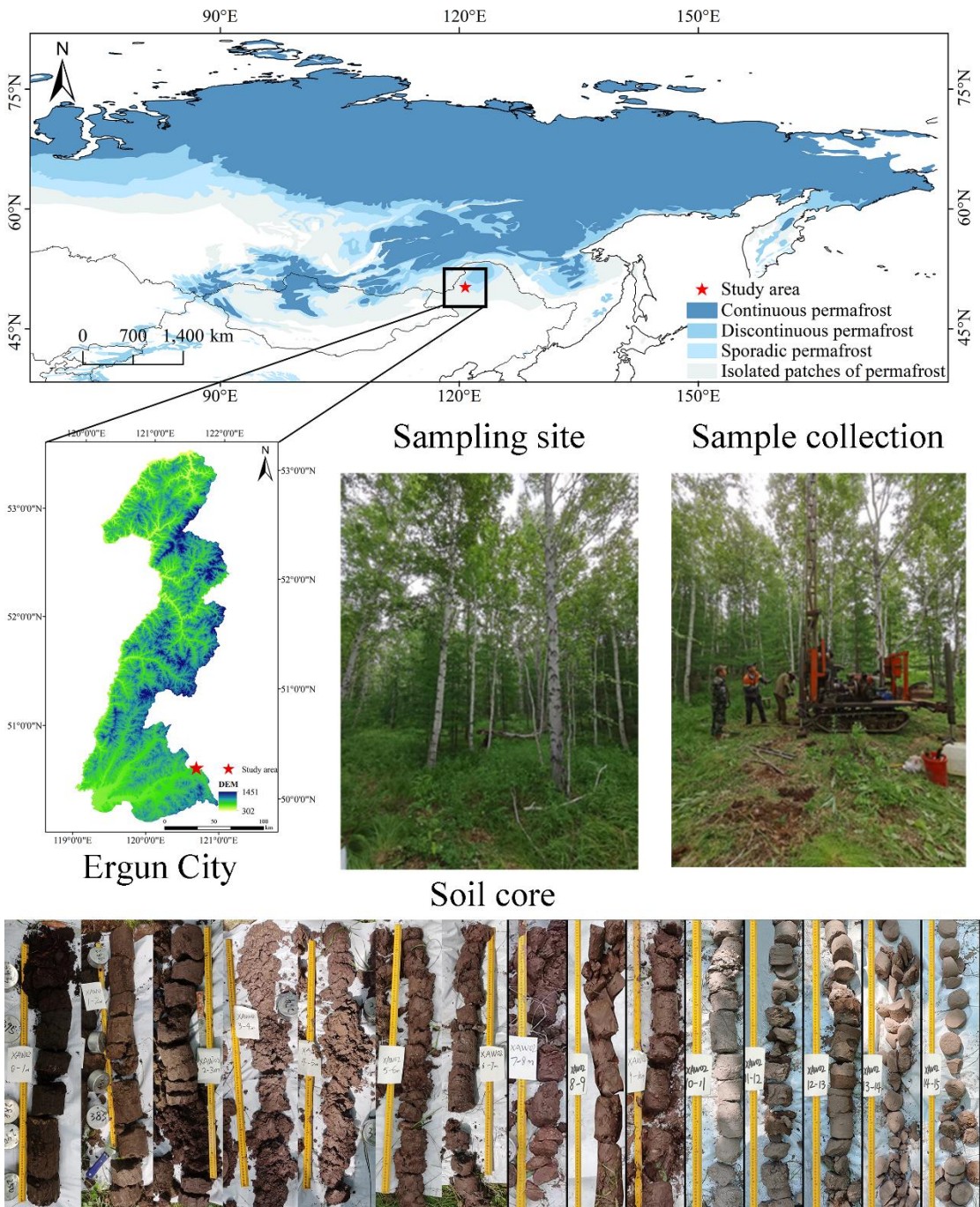

Figure 1. Study area and soil core sample collection. Permafrost distribution is adapted from Circum-Arctic map of permafrost and ground ice condition (Brown et al., 1997).

**2.2 Physicochemical analysis and spectral analysis of WSOC**

Gravimetric soil moisture (GSM) was quantified using the gravimetric method (Reynolds, 1970). Soil pH was measured with a PHS-3E pH meter (Leici, China) after shaking a soil-water suspension at a ratio of 1:2.5 (w/v). Soil electrical conductivity (EC) was determined using a DDSJ-319L conductivity meter (Leici, China) with a soil-to-water ratio of 1:5 (w/v). Soil organic carbon (SOC) and total carbon (TC) contents were determined by the dry combustion method using a Multi N/C 3100 analyzer (Jena,

Germany) (Nelson and Sommers, 1996); the TC and SOC samples were first air-dried. Before SOC
determination, approximately 100 mg of sample was weighed into a ceramic boat, an excess of 4 mol $L^{-1}$
HCl was added until no bubbles evolved, the mixture was thoroughly homogenized and left to stand
for 4 hours, and then dried at 65℃ for 16 hours prior to analysis. Soil inorganic carbon (SIC) was
calculated through differential subtraction.
Water-soluble organic carbon (WSOC) was extracted by adding fresh soil samples sieved through a 2
mm sieve, to deionized water at a ratio of 1:5 (w/v). The mixture was shaken continuously for 4 hours at
200 r $min^{-1}$ and 25°C. The samples were then centrifuged for 15 minutes at 4500 r $min^{-1}$ and filtered
through 0.45 μm glass fiber filters (Jones and Willett, 2006). A portion of the filtrate was used for
ultraviolet spectroscopic analysis, and the remaining filtrate was acidified by adding 3 mol $L^{-1}$
hydrochloric acid to adjust the pH to ≤2, effectively removing inorganic carbon. The pretreated samples
were stored at 4°C and analyzed within a week using the dry combustion method with a Multi N/C 3100
analyzer (Jena, Germany).
Total nitrogen (TN) was converted into ammonium nitrogen through an oxidation-reduction reaction
under the influence of concentrated sulfuric acid, sodium thiosulfate, and a catalyst (Kirk, 1950), and
quantified using the ammonia nitrogen module of a SAN++ flow injection auto-analyzer (Skalar,
Holland). Ammonium nitrogen ($NH_4^+$-N) and nitrate nitrogen ($NO_3^-$-N) were determined after extracting
2 g of fresh soil into 10 mL of 2 mol $L^{-1}$ potassium chloride solution, shaking for 2 hours at 200 r $min^{-1}$,
then centrifuging for 3 minutes at 8000 r/min and filtering through 0.45 μm glass fiber filters (Li et al.,

146     2012).

Total phosphorus (TP) in soil was determined using sodium hydroxide to convert all phosphorus-
containing minerals and organic phosphorus compounds into soluble orthophosphates (Sparks et al.,
2020), which were then quantified using a SAN++ flow injection auto-analyzer (Skalar, Holland).
Different WSOC compounds exhibit distinct spectral properties, and ultraviolet-visible (UV-Vis)
absorption spectra are commonly used to assess WSOC quality. The absorbance at 254 nm ($SUVA_{254}$) is
strongly correlated with WSOC aromaticity (Weishaar et al., 2003b). The *E250/E365* ratio, indicative of
WSOC aromaticity, humification degree, and molecular size (Helms et al., 2008), is also an important
parameter for characterizing WSOC. The absorbance values of WSOC at 250, 254, and 365 nm were
measured using a Lambda 35 UV/VIS spectrometer (PerkinElmer, USA) with a 10 mm quartz cuvette.

For each sample, the SUVA$_{254}$ value was calculated by dividing the UV absorbance measured at 254 nm by the WSOC concentration and multiplying 100 (Weishaar et al., 2003a). The *E250/E365* ratio was obtained by dividing the absorbance value at 250 nm by that at 365 nm (Helms et al., 2008). Throughout the incubation period, including the initial measurement on Day 0, UV-Vis spectroscopy was conducted on a portion of the WSOC extract to assess quality parameters such as SUVA$_{254}$ and the *E250/E365* ratio.

The SUVA$_{254}$ and the *E250/E365* ratio are widely used but semi-quantitative indicators. SUVA$_{254}$ can be inflated by non-aromatic UV-absorbers such as nitrate and dissolved Fe(III) and cannot distinguish among different aromatic moieties (Weishaar et al., 2003b; Logozzo et al., 2022). The *E250/E365* ratio provides only a coarse estimate of mean chromophore size; it is sensitive to baseline drift, light scattering, and becomes unreliable at low absorbance (Peuravuori and Pihlaja, 2004). By contrast, excitation–emission matrix (EEM) fluorescence spectroscopy yields multidimensional data with high sensitivity at low organic-matter concentrations (Anumol et al., 2015; Sgroi et al., 2018). In this study we therefore applied three-dimensional fluorescence spectroscopy to characterize water-extractable organic matter (WEOM). Fluorescence spectra were partitioned into five regions on the basis of integrated fluorescence area (Chen et al., 2003) (Table S1.). WEOM was extracted by adding deionized water to fresh soil samples sieved through a 2 mm sieve at a soil-to-water ratio of 1:5 (w/v), followed by shaking at 200 r min$^{-1}$ for 24 hours at 25°C. The samples were then centrifuged at 4500 r min$^{-1}$ for 15 minutes, and the supernatant was filtered through a 0.45 μm glass fiber filter to obtain WEOM (Zhou et al., 2023). A three-dimensional fluorescence spectrophotometer (Aqualog, HORIBA Scientific, France) was used to identify the fluorescent substances in water soluble organic matter. The excitation wavelength was set from 200 to 450 nm and the emission wavelength from 250 to 550 nm, with both the excitation and emission sampling intervals and slits adjusted to 5 nm, and the scanning speed maintained at 12,000 nm min$^{-1}$.

Because solute concentrations are lower in deeper-soil extracts, the extraction time for WEOM was extended relative to that for WSOC to obtain sufficient concentration and fluorescence signal (Zhou et al., 2023). Although a longer extraction can introduce minor compositional changes (Corvasce et al., 2006; Park and Snyder, 2018), EEM fluorescence nevertheless remains a robust method for assessing WSOC biodegradability (Vonk et al., 2015; Mu et al., 2017; Zhou et al., 2023).

**2.3 Laboratory Incubation experiment**

In the laboratory incubation experiments, we assessed the biodegradable water-soluble organic carbon

(BWSOC) at various soil depths over a period of 28 days, with measurements of WSOC content taken
on days 0, 2, 7, 14, and 28 (Vonk et al., 2015; Mu et al., 2017).
To minimize variability, WSOC samples were extracted in bulk from each soil layer. Fresh soil samples,
sieved through a 2 mm sieve, were mixed with deionized water at a soil-to-water ratio of 1:5 (w/v),
shaken continuously at 200 r min$^{-1}$ and 25°C for 4 hours, centrifuged at 4500 r min$^{-1}$ for 15 minutes, and
filtered through 0.45 μm filters. The resulting WSOC solution (500 mL) from each soil layer was
thoroughly homogenized. Aliquots of 30 mL of the homogenized WSOC solution were transferred into
50 mL sterile serum bottles.
To prepare the microbial inoculum, fresh soil samples from each soil layer were sieved through a 2
mm sieve to remove debris and large particles. The sieved soil was mixed with sterile deionized water at
a ratio of 1:5 (w/v) and shaken continuously at 200 r min$^{-1}$ and 25°C for 4 hours. This process facilitated
the release of microorganisms from soil particles into the aqueous phase, allowing them to enter the
extract in suspension form (Bottomley et al., 2020). The suspension was then centrifuged at 4500 r min$^{-1}$
for 15 minutes to remove any remaining soil particles. The supernatant was filtered through pre-
combusted (450°C for over 4 hours) Whatman GF/C filters with a pore size of 1.2 μm to obtain the
microbial inoculum. Finally, 3 mL of the inoculum (constituting 10% of the total volume) was added to
the water samples to introduce indigenous soil microorganisms from the respective soil depths (Vonk et
al., 2015). Inocula and WSOC samples were always matched by depth, preserving the natural microbe–
substrate association that is critical for realistic assessments of WSOC bioavailability and degradation
potential (Bhattacharyya et al., 2022; Pei et al., 2025).
To minimize nutrient limitations on microbial activity, standardized amounts of ammonium nitrate
($NH_4NO_3$) and dipotassium hydrogen phosphate ($K_2HPO_4$) were added to each sample. Specifically, a
0.02674 mol L$^{-1}$ $NH_4NO_3$ stock solution was prepared by dissolving 2.14 g of $NH_4NO_3$ in 1 L of
deionized water. Then, 100 μL of this stock solution was added to each 33 mL sample, resulting in final
concentrations of approximately 80 μmol L$^{-1}$ for $NH_4^+$ and $NO_3^-$. Similarly, a 0.0334 mol L$^{-1}$ $K_2HPO_4$
solution was prepared by dissolving 5.8176 g of $K_2HPO_4$ in 1 L of deionized water, which was
subsequently diluted tenfold to obtain a 0.00334 mol/L working solution. We added 100 μL of the diluted
$K_2HPO_4$ solution to each sample, achieving a final $PO_4^{3-}$ concentration of approximately 10 μmol L$^{-1}$
(Mu et al., 2017; Vonk et al., 2015). Previous studies suggested that these additions are sufficient to

prevent nutrient limitation and to standardize microbial activity (Mehring et al., 2013; Helton et al., 2015).
By equalizing nutrient supply across soil layers, we attribute any differences in WSOC consumption to
the intrinsic properties of the WSOC itself. Each sample was incubated in triplicate, along with two
control blanks: one with deionized water and another with deionized water plus nutrients, for a total of
five samples per depth interval. All samples were incubated at 20°C in the dark in a constant temperature
incubator (Thermo, USA), with caps partially opened. The samples were shaken once daily to maintain
aerobic conditions.

On measurement days, the samples were re-filtered through a 0.45 µm glass fiber filter to exclude
filterable microbial biomass. The quantified WSOC degradation accounted for both microbial
mineralization and assimilation processes. Part of the samples was immediately used for absorbance
measurements at wavelengths of 250 nm, 254 nm, and 365 nm. Another portion was acidified using 3
mol/L hydrochloric acid to adjust the pH to ≤2 and subsequently stored at 4°C, with WSOC concentration
measured within a week. BWSOC was determined by subtracting the WSOC content on day 28 from the
WSOC content on day 0. BWSOC (%) was calculated by dividing BWSOC by the WSOC content on
day 0 and multiplying by 100 %. The formulas for calculating BWSOC and BWSOC (%) are provided
in the Supporting Information. All experimental procedures were conducted on a sterile laminar flow
bench.

**2.4 Data analysis**

Pearson correlation analysis was used to explore the relationships between various environmental factors
and characteristics of WSOC. One-way ANOVA was employed to test the significant differences in the
molecular composition of WSOC, which was indicated by the $SUVA_{254}$ and $E250/E365$ ratios, across
different soil depth. To compare the differences in biodegradable water-soluble organic carbon (BWSOC)
across soil depths, the non-parametric Kruskal-Wallis test was applied. The biodegradability of water-
soluble organic carbon at time ($BWSOC_t$) was underwent nonlinear exponential fitting to obtain the
reaction kinetics constant ($k$). All statistical analyses were performed using R version 4.4.0
(https://www.r-project.org/).

**3 Results**

**3.1 Physicochemical properties**

242 The concentration of nutrients in the soil gradually decreases with depth (Fig. 2). In the surface layer (0-

243 30 cm), nitrogen content is lowest at the 10-20 cm depth. Electrical conductivity and total phosphorus

244 content are highest at 700-740 cm. The WSOC content ranged from 0.123 g kg$^{-1}$ to 0.355 g kg$^{-1}$. On

245 average, WSOC content was 0.246 g kg$^{-1}$ in the upper 0-2 m layer, while below 2 m it decreased to an

246 average of 0.183 g kg$^{-1}$. The highest WSOC content, at 0.355 g kg$^{-1}$, was observed between 160-180 cm.

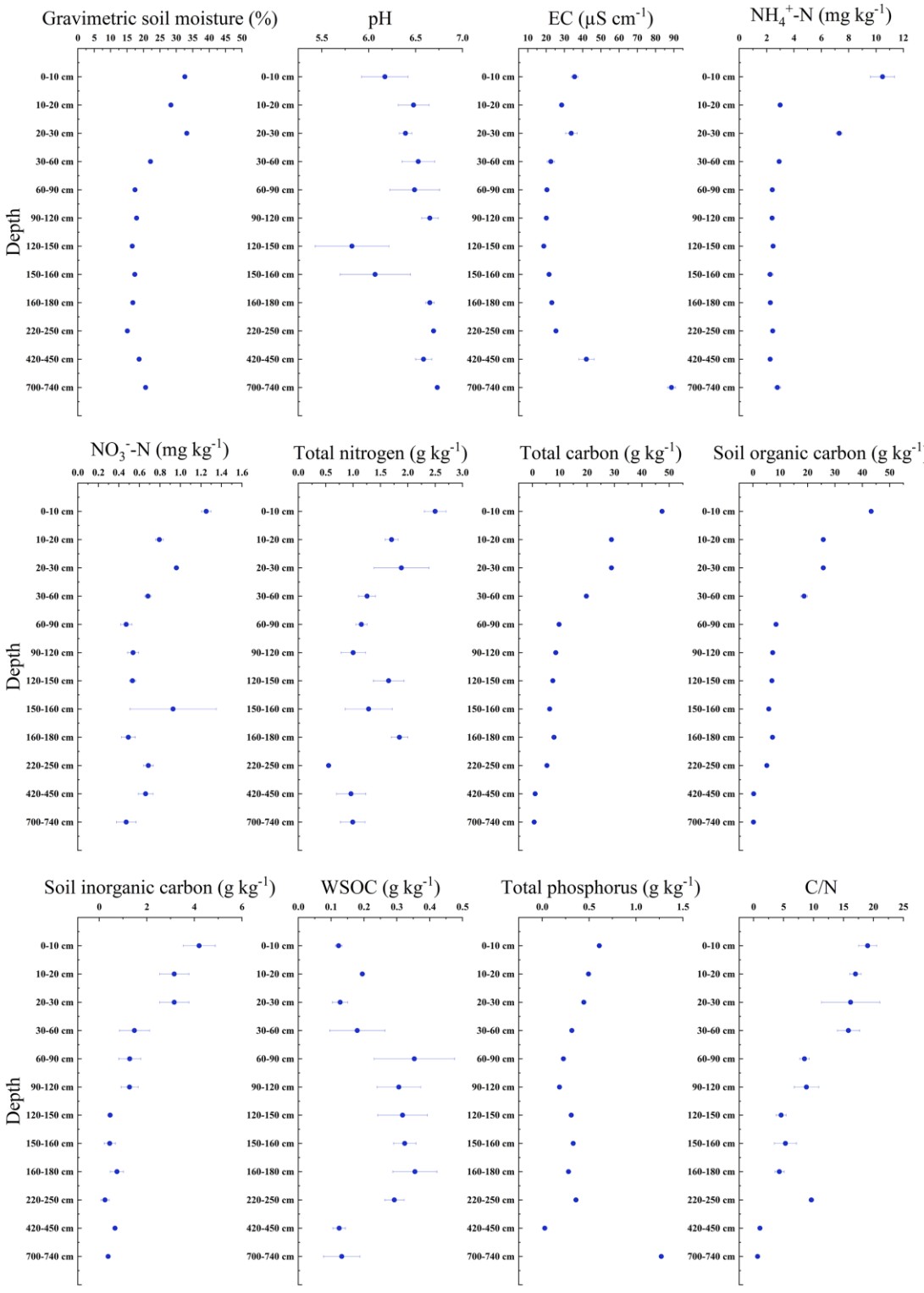

Figure 2. Soil physicochemical characteristics at different depth, error bars represent the standard error (n=3).

**3.2 Spectroscopy of water-soluble organic carbon**

There were significant differences in the aromaticity and molecular weight of WSOC between the 0-60

cm depth and deeper layers within the boreal forest ecosystem (n=3, $p<0.05$) (Fig. 3). The WSOC in the 0-60 cm depth predominantly consists of components with higher aromaticity and larger molecular weights. In contrast, deeper layers have WSOC with smaller molecular weights and less aromaticity (Fig. 3). Additionally, three-dimensional fluorescence spectroscopy displayed two major fluorescence peaks (Fig. 4): one in Region III, representing fulvic acid-like matter, and another in Region V, representing humic acid-like matter. The fluorescence intensity of fulvic acids is high across all depths, with significantly greater intensity in the 0-30 cm and 420-740 cm depths compared to other depths (Fig. 5).

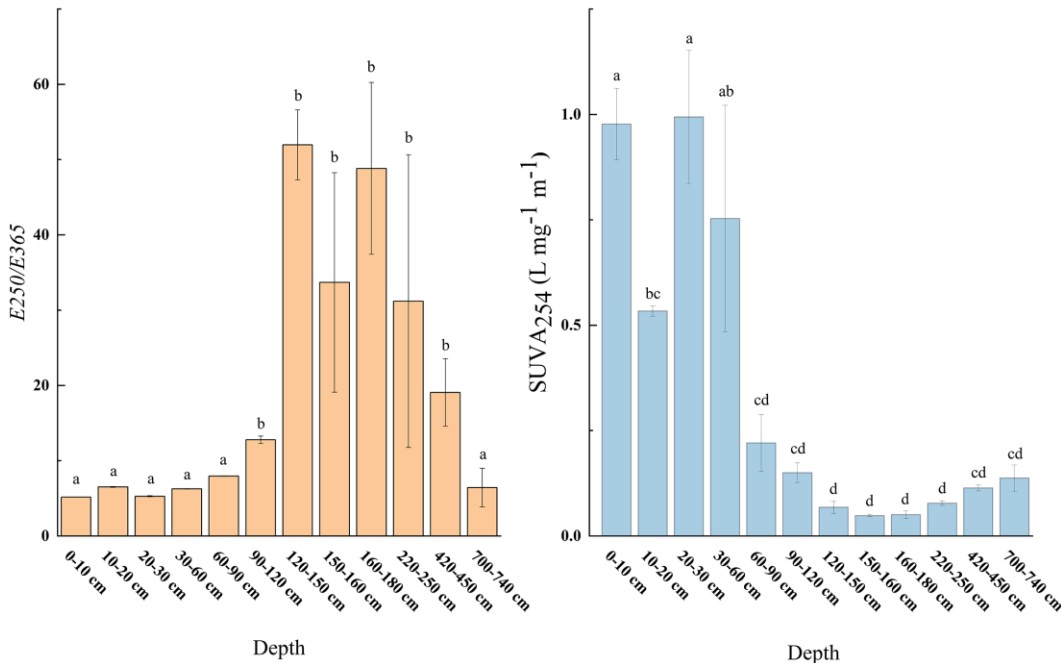

Figure 3. *E250/E365* and SUVA$_{254}$ at different depths. Different letters represent significant differences among different sampling points (n=3, $p < 0.05$), error bars represent the standard error.

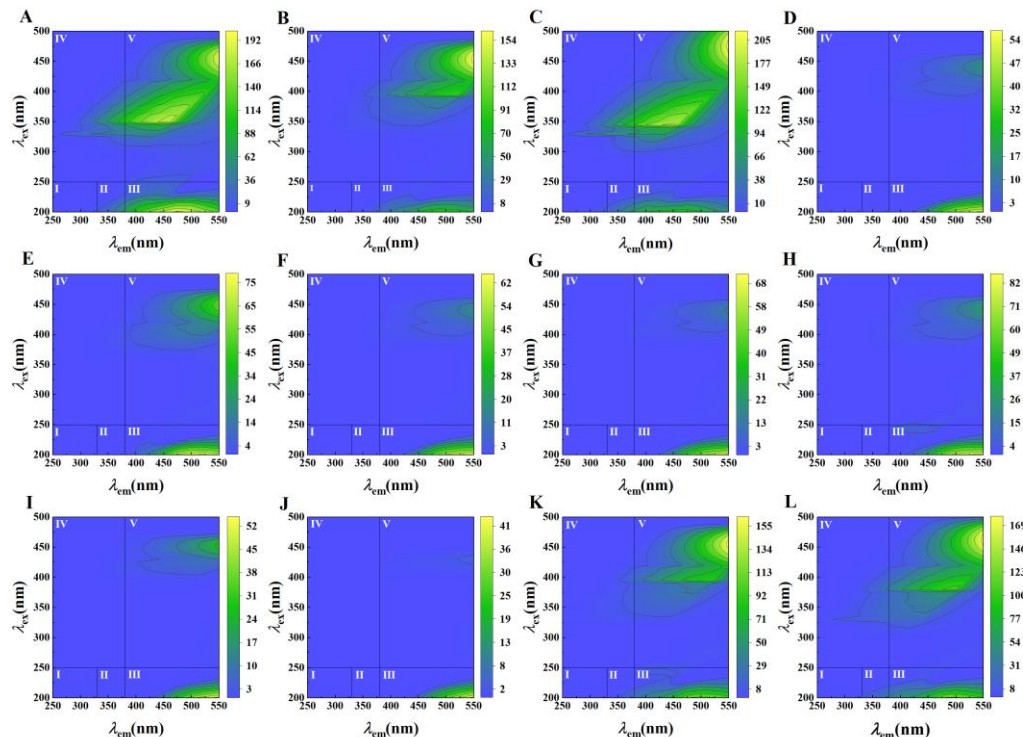


Figure 4. EEM fluorescence spectra of WSOM at different depths. Regions I, II, III, IV, and V are,
respectively, for tyrosine-like aromatic protein, tryptophan-like aromatic protein, fulvic acid-like matter,
soluble microbial byproduct-like matter, and humic acid-like matter. A: (0-10 cm); B: (10-20 cm); C:
(20-30 cm); D: (30-60 cm); E: (60-90 cm); F: (90-120 cm); G: (120-150 cm); H: (150-160 cm); I: (160-
180 cm); J: (220-250 cm); K: (420-450 cm); L: (700-740 cm).

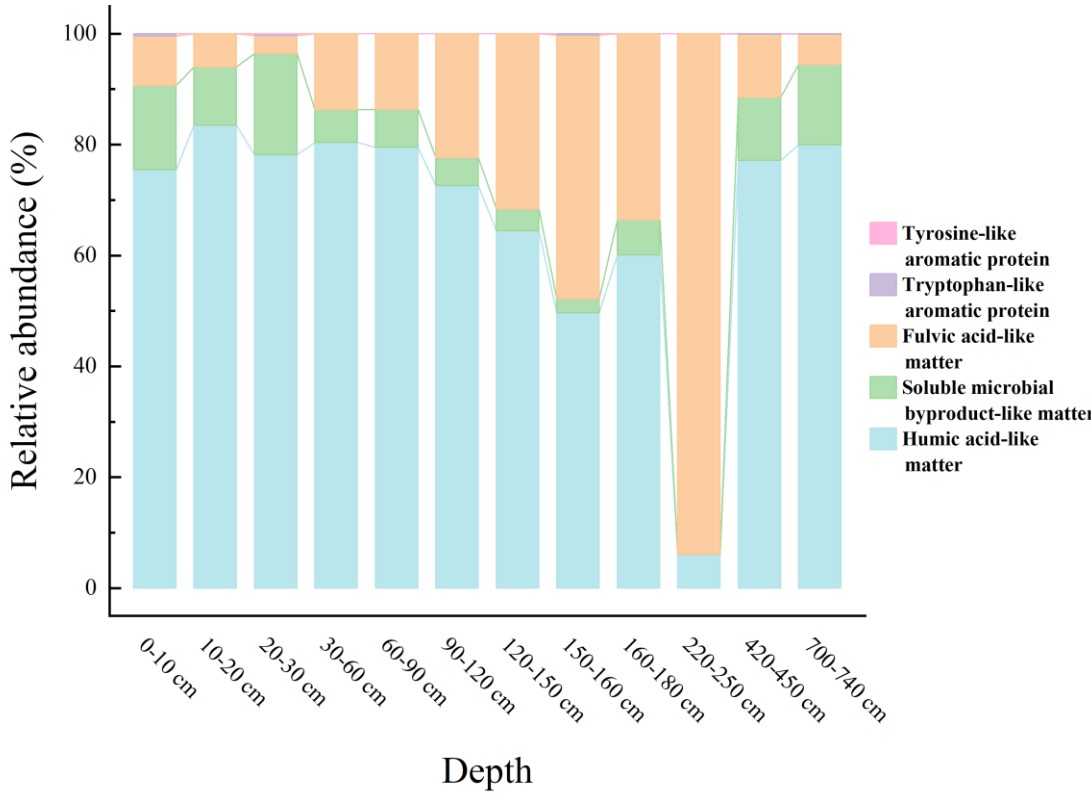


Figure 5. The EEM fluorescence spectra of WSOC at different depths
**3.3 Biodegradable water-soluble organic carbon, and the reaction kinetics constant *k***
BWSOC content and degradation kinetics exhibited pronounced, non-linear depth patterns (Fig. 6). Soils
at 60-180 cm depth exhibited higher BWSOC content and degradability compared to other depths (Fig.
6). Significant variations in degradation rates were observed during the incubation process. The reaction
kinetics constant (*k* values) indicated that WSOC degradation rates were lower in deeper soils (220-740
cm) (0.0681-0.0863 day$^{-1}$) (Table 2), mainly recorded between days 14 and 28 of incubation. In contrast,
the WSOC at 60-90 cm depth decomposed faster during the early stages of incubation, with a *k* value of
1.0952 (day$^{-1}$). Although deeper soils (below 2m) also contain relatively high BWSOC content,
decomposition in these layers occurred primarily during the later stages of incubation (days 14–28),
whereas the WSOC in upper layers (0-180 cm) was rapidly decomposed at the beginning of the
incubation period (Fig. 7).
**Table 2. Content of BWSOC, BWSOC (%), reaction kinetics constant (k), and coefficient of**
**determination (R2) at different soil depths.**

| Depth | BWSOC (g kg$^{-1}$) | BWSOC % | *k* (d$^{-1}$) | R$^2$ |
|---|---|---|---|---|

| | | | |
|---|---|---|---|
| 0-10 cm | 0.089±0.009 | 72. 96%±13.41% | 0.0497 | 0.9394 |
| 10-20 cm | 0.159±0.014 | 81.68%±9.18% | 0.4991 | 0.7484 |
| 20-30 cm | 0.127±0.011 | 90.43%±0.55% | 0.1302 | 0.8735 |
| 30-60 cm | 0.136±0.064 | 68.08%±3.79% | 0.3604 | 0.5532 |
| 60-90 cm | 0.321±0.098 | 91.67%±4.14% | 1.0952 | 0.9847 |
| 90-120 cm | 0.290±0.046 | 95.25%±4.98% | 0.3651 | 0.9360 |
| 120-150 cm | 0.285±0.052 | 90.45%±5.05% | 0.1394 | 0.8549 |
| 150-160 cm | 0.306±0.025 | 94.54%±2.32% | 0.0601 | 0.8823 |
| 160-180 cm | 0.311±0.040 | 88.21%±5.45% | 0.0737 | 0.9058 |
| 220-250 cm | 0.215±0.026 | 73.46%±1.31% | 0.0863 | 0.8910 |
| 420-450 cm | 0.101±0.017 | 80.66%±1.55% | 0.0712 | 0.9747 |
| 700-740 cm | 0.116±0.045 | 88.25%±2.81% | 0.0681 | 0.8692 |

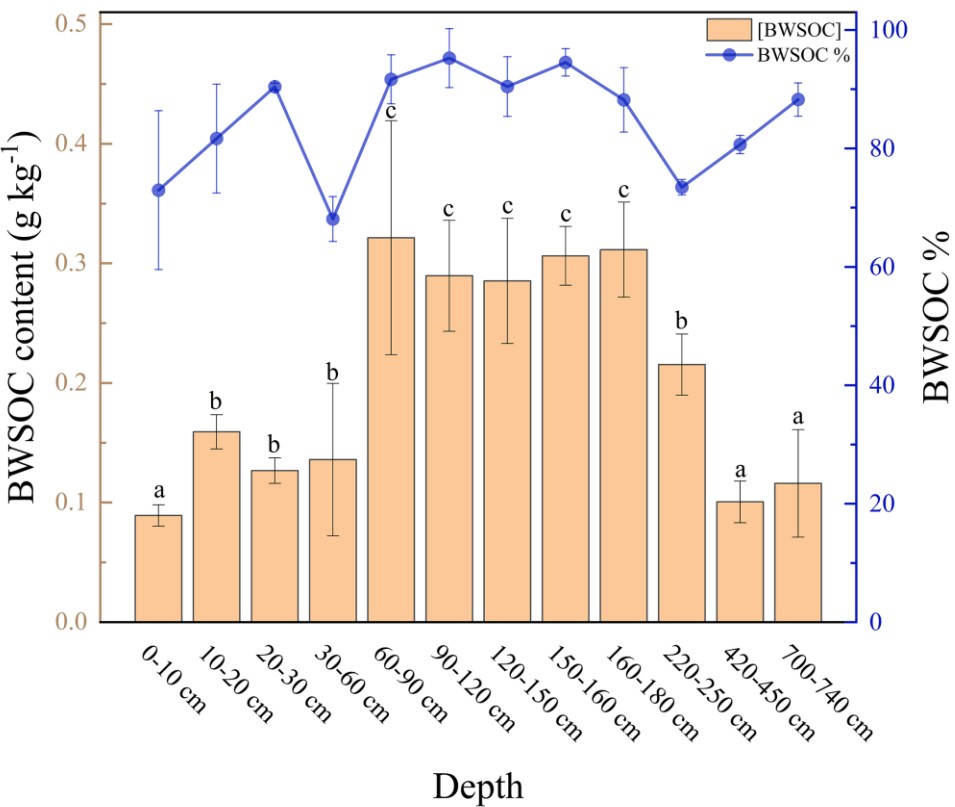


Figure. 6 Content of biodegradable water-soluble organic carbon (BWSOC) and the percentage of
biodegradable water-soluble organic carbon (BWSOC%) at different depths, error bars represent the
standard error (n=3).

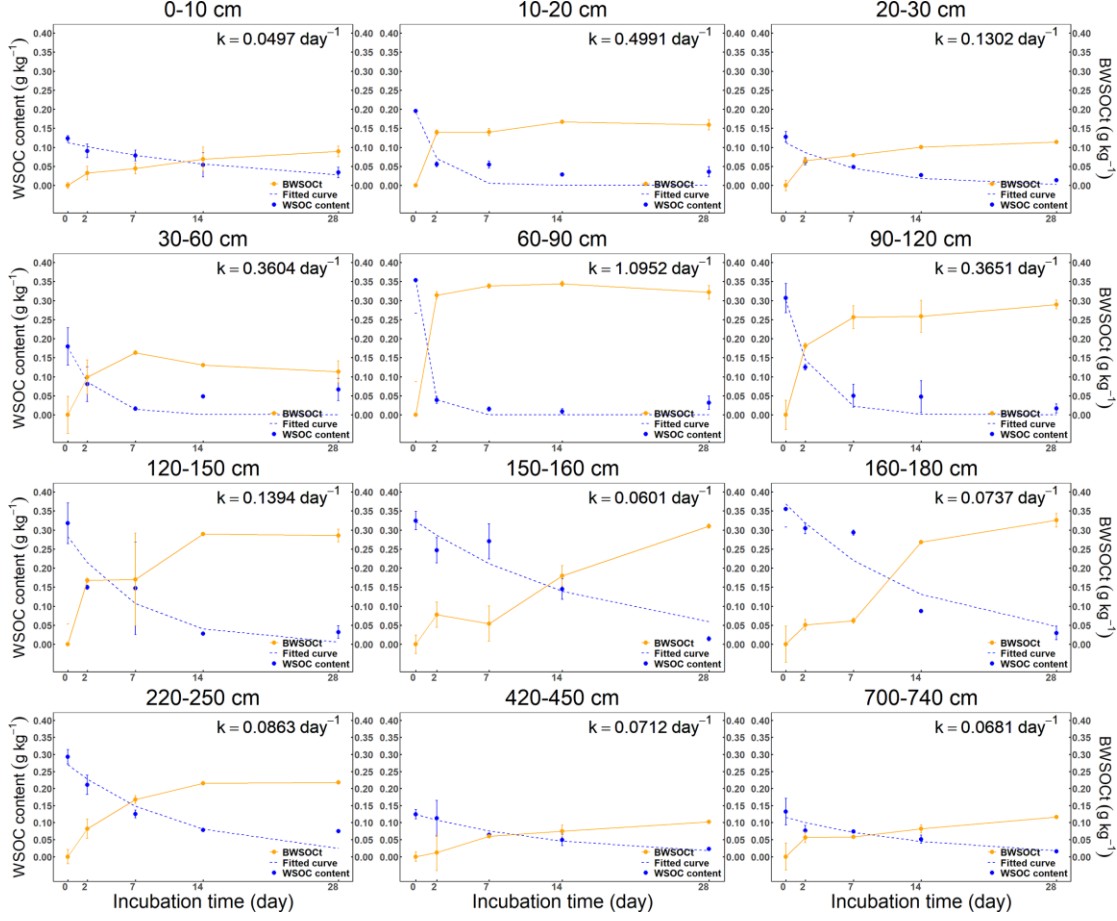

Figure 7. Water-soluble organic carbon content during the 28-day incubation at various depths. The blue curve is a nonlinear exponential fitting of WSOC content. The red curve illustrates the changes in biodegradable water-soluble organic carbon (BWSOC), with the *k*-value representing the reaction kinetics constant, error bars represent the standard error (n=3).

At 160-180 cm depth, $SUVA_{254}$ values gradually increased over the incubation period, while *E250/E365* value steadily decreases (Fig. 8). This indicates that as the incubation time increases, the aromaticity and molecular weight of the remaining WSOC also increase. In contrast, WSOC at other depths is rapidly decomposed during the initial stages of incubation, leading to a quick increase in $SUVA_{254}$ and a rapid decrease in *E250/E365* in the first 0-7 days, reflecting the rapid utilization of smaller, less aromatic molecules early in the incubation. The WSOC content at depths of 60-120 cm and the absorbance values at depths of 220-740 cm were extremely low by day 28, which likely resulted in very

low SUVA$_{254}$ and *E250/E365* values on that day. As a result, we excluded these data from the analysis.

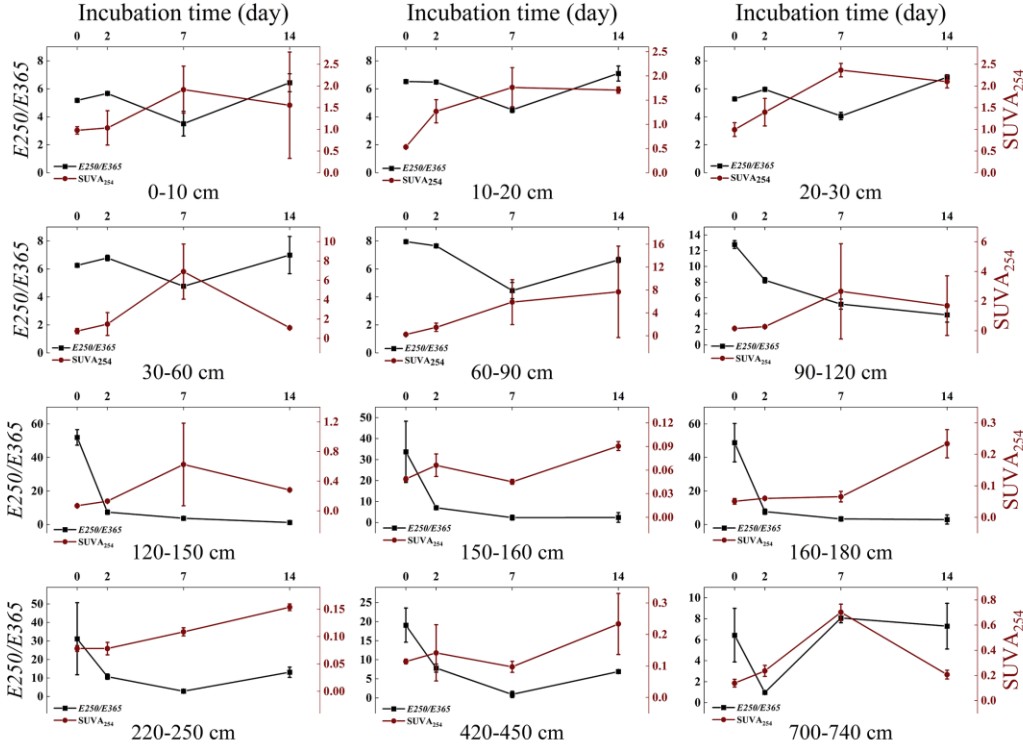

Figure 8. Water soluble organic carbon *E250/E365* and SUVA$_{254}$ during the 14 days of incubation at
different soil depths, error bars represent the standard error.

**3.4 Relationship among the biodegradation of water-soluble organic carbon and physicochemical**

**parameters**

Total carbon (TC), total nitrogen (TN), WSOC and its degradability showed significantly negative
correlations with depth. The aromaticity of WSOC (SUVA$_{254}$) and molecular weight (*E250/E365*)
showed significant correlations with biodegradable water-soluble organic carbon (BWSOC). *E250/E365*
showed a positive correlation with BWSOC (r = 0.528), while SUVA$_{254}$ was negatively correlated with
BWSOC (r = -0.582). Additionally, SUVA$_{254}$ and *E250/E365* demonstrated a strong negative correlation
(r = -0.589), suggesting that the molecular composition of WSOC significantly impacts its
biodegradability. The degradation rate (*k*) and the degree of biodegradability (BWSOC %) of WSOC has
no significant correlations with other physicochemical parameters (Fig. 9).

313         Simple-linear regressions showed that BWSOC declined with SUVA$_{254}$ ($\beta$ = -0.158 ± 0.062, p = 0.029,

$R^2$ = 0.395) and increased with *E250/E365* ($\beta$ = 0.003 ± 0.001, p = 0.035, $R^2$ = 0.374), whereas the
degradation constant *k* was not significantly related to either index (p ≥ 0.05) (Table S2., Fig. S1). A
multiple model including both optical variables accounted for 35 % of the variance in BWSOC (adjusted

R$^2$ = 0.350, p = 0.058) but remained non-significant for k (adjusted R$^2$ = 0.043) (Table S3 and S4., Fig. S2).

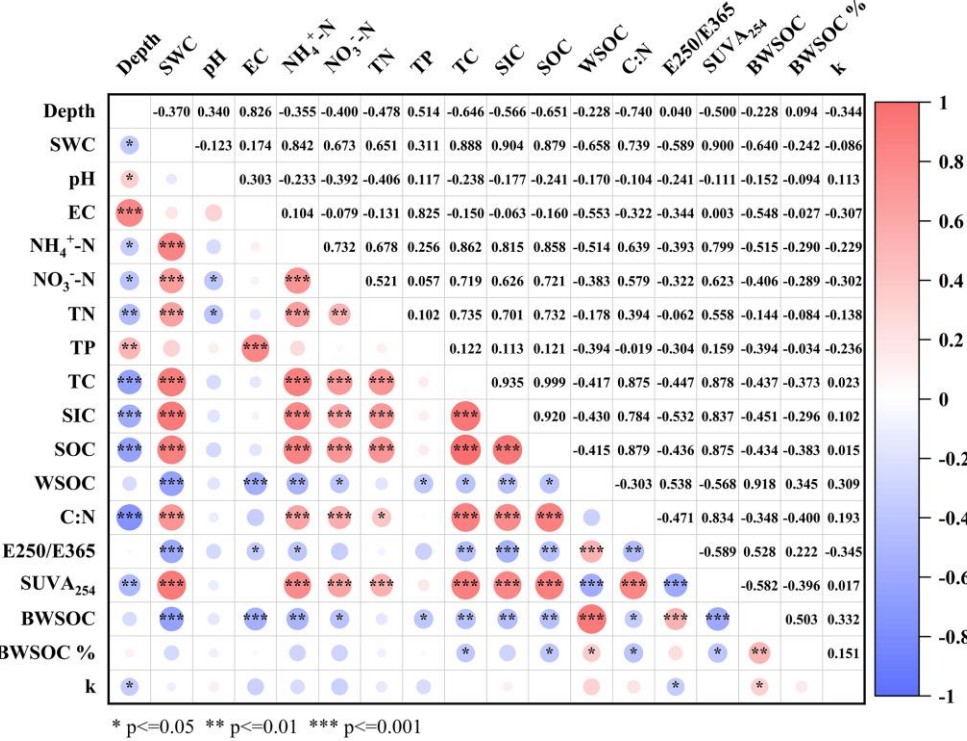

Figure 9. Correlation coefficients among different environmental factors (n=12). Red indicates a positive correlation, while blue indicates a negative correlation. The deeper the color, the stronger the correlation. The color gradient ranges from -1 (complete negative correlation) to +1 (complete positive correlation).

## 4 Discussion

### 4.1 Water-soluble organic carbon content and spectral signature

At depths of 60–180 cm, WSOC concentrations were relatively higher than in other soil layers. This pattern is plausible since higher organic matter inputs from roots and litter generally occur in these upper soil layers, facilitating WSOC accumulation (Hu et al., 2014). Additionally, the silty clay loam texture at these depths contains a substantial proportion of silt and clay particles, creating a denser pore structure capable of effectively adsorbing and retaining organic matter (Bucka et al., 2023). With increasing soil depth, the higher sand content can lead to lower porosity but higher macroporosity (Mentges et al., 2016). Consequently, higher sand content reduces the potential for SOC preservation (Bucka et al., 2023), resulting in lower WSOC concentrations in deeper layers.

We found that the WSOC in the 0-60 cm soil layer exhibited stronger aromaticity and larger molecular weight. Three-dimensional fluorescence spectroscopy confirmed that the WSOC in the surface layer was primarily composed of larger molecular humic substances, which aligns with previous findings from boreal forests in Alaska (Wickland et al., 2007). These substances are primarily plant-derived (Walker et al., 2013; Mann et al., 2016) and are often associated with root exudates and microbial exometabolites abundant in the upper soil horizons (Raudina et al., 2017).

Deep soils exhibited a higher proportion of fulvic-like substances in their WSOC, characterized by smaller molecular weights and lower aromaticity. This finding suggests that WSOC in deeper layers generally possesses lower aromaticity and molecular weight (Fouché et al., 2020). Although SOC in deep soils is usually considered has high fresh organic materials due to the low temperature limits the microbial decomposition (Heffernan et al., 2024), our results suggest that long term accumulation of highly decomposed organic matter that forms low-molecular-weight fulvic acid-like substances with lower aromaticity is still abundant in deep soils (Corvasce et al., 2006; Lv et al., 2020). This mechanism helps explain the observed decrease in WSOC aromaticity and molecular weight with increasing soil depth (Koven et al., 2015; Panneer Selvam et al., 2017; Drake et al., 2015).

**4.2 Biodegradable water-soluble organic carbon, and the reaction kinetics constant *k***

Water-soluble organic carbon (WSOC) in boreal forest soils is highly biodegradable, with the largest proportion of biodegradable WSOC consistently occurring at depths of 60-160 cm. This depth-dependent pattern was reproduced in all replicates. In this study, the 60-160 cm interval has a buried organic horizon that contains high organic-matter concentrations (Werdin-Pfisterer et al., 2012). Spectroscopic results further confirm that the depth-related differences in BWSOC % arise from variations in the chemical composition of water-extractable organic matter. In Alaskan Kolyma River basin, WSOC concentrations decreased by about 50% following a seven-day incubation (Spencer et al., 2015); in deep soils, ancient low-molecular-weight organic acids within WSOC are rapidly mineralized, leading to a ~53% decline in WSOC after 200 hours of incubation (Koven et al., 2015). The high biodegradability of WSOC is closely related to its chemical composition (Burd et al., 2020). In these regions, WSOC primarily consists of low-aromaticity, low-molecular-weight organic matter that is readily decomposed by microbes (Drake et al., 2015), making it easily accessible for microbial utilization (Ward and Cory, 2015).

Despite the high biodegradability of WSOC, decomposition rates in the deeper soils (220-740 cm)

were slower than those in the upper layers (0-180 cm), particularly during the later stages of incubation. This pattern suggests that microbes rapidly consumed the most bioavailable compounds in the deeper layers at the beginning of the incubation (Wild et al., 2014). Over time, the WSOC became increasingly aromatic, indicating that microbes had preferentially utilized the more easily decomposable organic matter early on (Drake et al., 2015). At 60–90 cm, WSOC had lower aromaticity and molecular weight than at 0–60 cm, contributing to faster degradation (Kalbitz et al., 2003a), particularly during the first 48 hours (Roehm et al., 2009). However, degradation rates declined with depth, likely because microbial abundance and activity also were lower in deeper horizons (Marschner and Kalbitz, 2003; Neff and Asner, 2001; Yano et al., 2000).

Our study highlights the differences in the biodegradability of WSOC at various soil depths in boreal forest ecosystems. However, it is important to note that the high values of biodegradable WSOC (BWSOC) and BWSOC (%) observed in this study may be influenced by several methodological factors (Dutta et al., 2006; Vonk et al., 2015; Abbott et al., 2014; Kaplan and Newbold, 1995; Frías et al., 1995). In our study, nutrient amendments were used, and the samples were incubated under aerobic conditions at a constant temperature of 20°C in the dark. Uniform nutrient supply could have induced nutrient-saturation effects in the 0–60 cm samples, where the C/N ratio is likely close to the Redfield ratio, thereby lowering BWSOC % (Aber, 1992; Aber et al., 1997; Gress et al., 2007). Moreover, we used depth-matched inocula to preserve native microbe–substrate interactions, consistent with previous WSOC-biodegradability studies (Bhattacharyya et al., 2022; Pei et al., 2025; Vonk et al., 2015), while we did not quantify microbial abundance or community composition. Therefore, the high BWSOC % values reflect potential rather than in-situ decomposition rates. Future work is required to examine in-situ nutrient status, microbial mass and microbial community structure to better understanding the depth-dependent WSOC dynamics.

**4.3 Water-soluble organic carbon biodegradation and physicochemical parameters**

BWSOC in this study showed a negative correlation with physicochemical parameters. Similar negative EC–BWSOC patterns have been reported for coastal wetland soils, where rising salinity restrained microbial residue accumulation and SOC turnover (Qu et al., 2018; Shao et al., 2022b), and laboratory studies suggest that osmotic stress can curb microbial respiration of labile dissolved organic carbon (Yang et al., 2018). Nutrient effects are strongly context dependent; for instance, nitrogen enrichment reduced

soil-carbon mineralization in incubation experiments with cropland and grassland soils (Perveen et al., 2019), whereas field fertilization in a boreal forest increased DOC concentrations under nitrate addition (Shi et al., 2019). These contrasting findings indicate that multiple, site-specific processes including osmotic stress, stoichiometric imbalance, shifts in microbial community composition, or sorption dynamics may underlie the correlations observed in this study. Further manipulation experiments are required to disentangle these mechanisms.

A significant correlation was also observed between BWSOC and both SUVA$_{254}$ and the *E250/E365* ratio. These results highlight the importance of WSOC properties in determining its biodegradability (Kalbitz et al., 2003b; Fellman et al., 2008). The composition of WSOC is influenced by physicochemical parameters such as total carbon, total nitrogen, total phosphorus, and pH (Li et al., 2018; Roth et al., 2019). The strong positive correlation between these physicochemical parameters and SUVA$_{254}$ and *E250/E365* can be attributed to the high concentration of nutrients, which promotes the accumulation and transformation of organic matter, leading to the formation of more complex and recalcitrant organic compounds (Takaki et al., 2022).

**5 Conclusion**

This study quantitatively analyzed the biodegradability of water-soluble organic carbon (WSOC) at various depths in a boreal forest. Our results show that BWSOC content ranges from 0.089 g kg$^{-1}$ to 0.321 g kg$^{-1}$, with the lowest observed biodegradability in surface soil WSOC still reaching 68.08%. Three-dimensional fluorescence spectroscopy indicated that surface WEOM is dominated by highly aromatic, humic-acid-like matter. With increasing depth, the proportion of fulvic-acid-like compounds rose, whereas WSOC aromaticity and molecular weight declined. As a result, biodegradability in soils below 2 m reached 80.8 %. Although WSOC in deep horizons degraded more slowly than in the upper profile, it remained highly biodegradable. Correlation analyses further indicate that the molecular composition of WSOC is a key factor influencing its biodegradability. Overall, WSOC contents at the southern margin of the boreal forest were comparable to those reported at higher latitudes. Because WSOC represents the most labile fraction of the soil organic carbon pool, our results suggest that continued climate warming could accelerate losses of labile SOC throughout the soil profile in boreal forests.

**Acknowledgements**

We sincerely thank Defu Zou, Guojie Hu, and their colleagues for their invaluable assistance with sample collection.

**Author contributions**

**YZ:** Conceptualization, Formal analysis, Investigation, Methodology, Writing – original draft. **CL:** Supervision, Resources. **RL:** Resources. **HW:** Validation, Resources. **XW:** Resources, Funding acquisition. **ZZ:** Investigation. **SZ:** Project administration, Data curation, Funding acquisition. **XW:** Writing – review and editing, Validation, Project administration.

**Funding sources**

The Science & Technology Fundamental Resources Investigation Program (2022FY100701), National Natural Science Foundation of China (U20A2082, 42430412, 32061143032), Basic scientific research business expenses of colleges and universities in Heilongjiang Province (2022KYYWF0181), Harbin Normal University Postgraduate Innovation Program (HSDSSCX2022108)

**Declaration of Competing Interest**

The authors declare that they have no known competing financial interests or personal relationships that could have appeared to influence the work reported in this paper.

**Data availability**

Data will be made available on request.

**Appendix A. Supplementary data**

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
