# Peer review of "High biodegradability of water-soluble organic carbon in soils at the southern margin of the boreal forest"

_EGUsphere, 2025_

## Author Comment (AC2)

Referee 1

General Comments

This manuscript investigates the content, molecular composition, and biodegradability of water-soluble organic carbon (WSOC) across a 7.4 m soil profile at the southern boundary of the boreal forest. The authors employ a combination of spectroscopic analyses and laboratory incubations to evaluate depth-resolved changes in WSOC properties and their relation to biodegradability. The study is timely and relevant given the sensitivity of boreal carbon stocks to climate change and permafrost degradation. It contributes valuable data on deep soil carbon dynamics, particularly from a region (southern margin of the boreal forest in Northeast China) that is underrepresented in current literature.

The manuscript is generally well-structured and clearly written, with a sound experimental design and appropriate data analysis. However, a few aspects require clarification or improvement to enhance the manuscript's scientific rigor and presentation. These include a better contextualization of the study's novelty, clearer methodological justifications, and refinement of certain interpretations to avoid potential overgeneralization.

Response: We sincerely thank the reviewer for their positive evaluation of our work and their constructive suggestions. We appreciate the recognition of our study's relevance and contribution to soil organic carbon research, especially in the underrepresented region of the southern boreal forest margin in Northeast China. We have carefully considered all comments and have revised the manuscript accordingly to improve its clarity, rigor, and presentation. Below, we provide point-by-point responses to the specific comments.

Specific Comments

Novelty and Contribution:

The manuscript would benefit from a clearer articulation of how this study advances current understanding beyond existing work (e.g., studies from Alaska, Siberia). The novelty lies in deep-profile in situ WSOC characterization at the southern boreal margin—this should be emphasized more explicitly in both the abstract and introduction.

Response: We appreciate the valuable comments and fully agree on the need to further highlight the novelty of this study. We added the following text in the revised version:

Abstract: In the abstract, we now emphasize that our innovation lies in the deep-profile in-situ WSOC characterization at the southern boreal margin. The revised sentence reads:

"Water-soluble organic carbon (WSOC) is an important component of the soil organic carbon pool. While the biodegradability and its compositional changes of WSOC in deep soils in boreal forests remain unknown. Here, based on spectroscopic techniques, we conducted a 28-day laboratory incubation to analyze the molecular composition, biodegradability, and compositional changes of WSOC during a laboratory incubation for deep soils at the southern boreal margin. The results showed that in the upper 2 m soils, the average content of biodegradable WSOC was 0.228 g/kg with an average proportion of 86.41% in the total WSOC. In the soil layer between 2.0-7.4 m, the average biodegradable WSOC content was 0.144 g/kg, accounting for 80.79% of the total WSOC. Spectroscopic analysis indicates that the WSOC in the upper soils is primarily composed of highly aromatic humic acid-like matter with larger molecular weights than those in deep soils. Both the aromaticity and molecular weight decrease with depth, and the WSOC is mainly composed of fulvic acid-like matter in the deep soils, suggesting high biodegradability of WSOC in the deep soils. Overall, our results suggest that the water-soluble organic carbon in the boreal forests exhibits high biodegradability both in the shallow layer and deep soils."

Introduction: We added a section explaining how this study advances current understanding beyond existing work (e.g., studies from Alaska and Siberia). The revised text is as follows:

Previous studies in permafrost regions showed that several factors can significantly influence the concentration, aromaticity, molecular weight, and optical characteristics of dissolved organic matter (DOM) (Kurashev et al., 2024). For instance, a freeze–thaw manipulation in a continuous permafrost region of northern Sweden showed that WSOC biodegradability increased as the freezing front deepened, largely because protein-like compounds accumulated at this depth (Panneer Selvam et al., 2016). The chemical nature of WEOM can be an important factor affecting the decomposition of SOM (Paré and Bedard-Haughn, 2013). In addition, hydrological-redox status can jointly control the stability of SOM (Pengerud et al., 2013). The tabular ground ice contains a high proportion of labile DOC that may accelerate the decomposition of permafrost SOM during melting (Semenov et al., 2024). These studies improved our understandings of DOM in permafrost regions, while few studies have been conducted in the southern boundary area of boreal forest, which may represent the future conditions of vast boreal forests due to the climate warming.

Many studies have been conducted to reveal the SOM characteristics within 3 m soils. In a 0–3 m permafrost profile in the Kolyma River Basin in Siberia, it was found that SOM in permafrost contain more water-soluble substrates and, after thaw, can be rapidly degraded by active microbes (Uhlířová et al., 2007). In a Northeast Siberia area, the active layer in around 60 cm, and it was found that the SOM from permafrost within 1 m depth was more sensitive to temperature changes than that of active layer (Walz et al., 2017). Since soil deep than 3 m in permafrost regions constitute a large proportion of permafrost carbon pools, and this carbon pool may also contribute to the future soil organic carbon cycle (Schuur et al., 2022), it is necessary to understand the SOM dynamics deeper than 3 m depth.

Microorganisms play a key role in the carbon cycle and strongly influence the biodegradability of WSOC (Marschner and Kalbitz, 2003; Kalbitz et al., 2003; Neff and Asner, 2001; Yano et al., 2000). Microbial biomass is more abundant in surface horizons, where soil-organic-carbon mineralization proceeds rapidly (Henneron et al., 2022; Pei et al., 2025), whereas thaw-activated bacteria in deeper layers can rapidly mineralize WSOC after permafrost thaws (Drake et al., 2015). Microbial use of WSOC is modulated by environmental factors such as soil

moisture (Zhang et al., 2024; Li et al., 2020) and soil physicochemical properties (Lv et al., 2024; Shao et al., 2022a). Therefore, detailed knowledge of the content, chemical composition, and biodegradability of WSOC along deep soil profiles is critical for clarifying how subsurface carbon is mobilized, transformed, and ultimately influences carbon cycling in boreal forests. The objective of this study is to quantify WSOC of soil profile that deeper than 3 m in a southern boreal margin. We conducted laboratory incubation experiments to determine differences in biodegradable water-soluble organic carbon (BWSOC) and employed spectroscopic techniques to reveal its compositional characteristics (Kothawala et al., 2014; Chavez-Vergara et al., 2014; Sun et al., 2022; Murphy et al., 2008; He et al., 2023). The results can improve our understandings of SOC in boreal forests under a warming climate.

Methodological Justification:

The use of nutrient amendments ($NH_4NO_3$ and $K_2HPO_4$) in biodegradability assays should be more critically discussed in the Methods and Discussion sections. While this standardizes microbial activity, it may inflate BWSOC estimates compared to natural field conditions.

Response: We appreciate the reviewer's insight. Although nutrient amendments are often used to standardize microbial activity, they may indeed lead to higher BWSOC estimates under laboratory conditions relative to natural conditions. We therefore expanded both the Methods and Discussion sections to provide a more critical account of this limitation.

Additions to the Methods section:

To minimize nutrient limitations on microbial activity, standardized amounts of ammonium nitrate ($NH_4NO_3$) and dipotassium hydrogen phosphate ($K_2HPO_4$) were added to each sample. Specifically, a 0.02674 mol/L $NH_4NO_3$ stock solution was prepared by dissolving 2.14 g of $NH_4NO_3$ in 1 L of deionized water. Then, 100 μL of this stock solution was added to each 33 mL sample, resulting in final concentrations of approximately 80 μmol/L for $NH_4^+$ and $NO_3^-$. Similarly, a 0.0334 mol/L $K_2HPO_4$ solution was prepared by dissolving 5.8176 g of $K_2HPO_4$ in

1 L of deionized water, which was subsequently diluted tenfold to obtain a 0.00334 mol/L working solution. We added 100 μL of the diluted $K_2HPO_4$ solution to each sample, achieving a final $PO_4^{3-}$ concentration of approximately 10 μmol/L (Mu et al., 2017; Vonk et al., 2015). Previous studies suggested that these additions are sufficient to prevent nutrient limitation and to standardize microbial activity (Mehring et al., 2013; Helton et al., 2015). By equalizing nutrient supply across soil layers, we attribute any differences in WSOC consumption to the intrinsic properties of the WSOC itself. Each sample was incubated in triplicate, along with two control blanks: one with deionized water and another with deionized water plus nutrients, for a total of five samples per depth interval. All samples were incubated at 20°C in the dark in a constant temperature incubator (Thermo, USA), with caps partially opened. The samples were shaken once daily to maintain aerobic conditions.

Additions to the Discussion section:

Our study highlights the differences in the biodegradability of WSOC at various soil depths in boreal forest ecosystems. However, it is important to note that the high values of biodegradable WSOC (BWSOC) and BWSOC (%) observed in this study may be influenced by several methodological factors (Dutta et al., 2006; Vonk et al., 2015; Abbott et al., 2014; Kaplan and Newbold, 1995; Frías et al., 1995). In our study, nutrient amendments were used, and the samples were incubated under aerobic conditions at a constant temperature of 20°C in the dark. Uniform nutrient supply could have induced nutrient-saturation effects in the 0–60 cm samples, where the C/N ratio is likely close to the Redfield ratio, thereby lowering BWSOC % (Aber, 1992; Aber et al., 1997; Gress et al., 2007). Moreover, we used depth-matched inocula to preserve native microbe – substrate interactions, consistent with previous WSOC-biodegradability studies (Bhattacharyya et al., 2022; Pei et al., 2025; Vonk et al., 2015), while we did not quantify microbial abundance or community composition. Therefore, the high BWSOC % values reflect potential rather than in-situ decomposition rates. Future work is required to examine in-situ nutrient status, microbial mass and microbial community structure to better understanding the depth-dependent WSOC dynamics.

The use of extracted microbial inoculum per soil layer could introduce variability due to differential microbial biomass and viability. Were microbial abundance or composition controlled or measured?

Response: We appreciate the reviewer's concern that depth-specific inocula could introduce variability. In this study, each horizon received the same fixed volume of slurry inoculum, following procedures described in earlier work (Vonk et al., 2015). Our primary objective was to quantify depth-dependent differences in the in-situ biodegradability of WSOC rather than to characterize microbial communities; therefore, we did not measure microbial biomass or community composition at each depth. We recognize that undetected variation in microbial abundance or viability may still have influenced the results. This limitation is now noted in the revised Discussion, and future work will include direct assessments of microbial biomass (e.g., PLFA, qPCR) and community structure (amplicon sequencing) to better disentangle biological from chemical controls on WSOC degradation. The revised discussion is as follows:

Moreover, we used depth-matched inocula to preserve native microbe–substrate interactions, consistent with previous WSOC-biodegradability studies (Bhattacharyya et al., 2022; Pei et al., 2025; Vonk et al., 2015), while we did not quantify microbial abundance or community composition. Therefore, the high BWSOC % values reflect potential rather than in-situ decomposition rates. Future work including direct assessment of microbial mass and community structure is required to examine in-situ nutrient status, microbial mass and microbial community structure to better understanding the depth-dependent WSOC dynamics.

Depth Resolution:

The authors group certain depths for discussion (e.g., "upper 2 m" vs. "below 2 m"), but patterns are often non-linear across depths. Consider incorporating a more nuanced, depth-wise interpretation where appropriate (especially for Fig. 6 and Table 2).

Response: We appreciate the reviewer's comment. In the revised manuscript we have (1) elaborated the Results to give a finer-scale explanation of depth-related patterns in WSOC properties and (2) expanded the Discussion to address their more nuanced, non-linear variations—especially for Fig. 6 and Table 2.

Revisions in the Results section:

[revised manuscript text omitted]

we did not quantify microbial abundance or community composition. Therefore, the high BWSOC % values reflect potential rather than in-situ decomposition rates. Future work is required to examine in-situ nutrient status, microbial mass and microbial community structure to better understanding the depth-dependent WSOC dynamics.

The high BWSOC% at 60–180 cm is striking—please clarify if this is a consistent finding or potentially due to sampling or incubation artifacts.

Response:  We have already clarified this in the Discussion:

Water-soluble organic carbon (WSOC) in boreal forest soils is highly biodegradable, with the largest proportion of biodegradable WSOC consistently occurring at depths of 60-160 cm. This depth-dependent pattern was reproduced in all replicates and, to the best of our knowledge, cannot be attributed to incubation or sampling artefacts. The 60-160 cm interval often coincides with a buried organic horizon that contains elevated organic-matter concentrations (Werdin-Pfisterer et al., 2012). Similar enrichments in readily mineralizable WSOC have been reported in Yedoma permafrost soils (Strauss et al., 2015) and have been linked to higher biodegradation rates during incubation experiments (Heslop et al., 2019). Spectroscopic results further confirm that the depth-related differences in BWSOC % arise from variations in the chemical composition of water-extractable organic matter.

Spectroscopic Interpretation:

The authors use $SUVA_{254}$ and *E250/E365* as proxies for aromaticity and molecular weight, respectively. This is acceptable, but readers would benefit from a brief discussion on their limitations and the role of fluorescence indices (e.g., FI, HIX) that were not used.

Response: We appreciate the suggestion to include fluorescence indices such as the Fluorescence Index (FI) and Humification Index (HIX). Unfortunately, the spectro-fluorometer settings chosen for this study were designed for rapid, qualitative EEM screening and do not

allow a reliable calculation of FI or HIX. Therefore, these indices are not reported here. We recognize their value for assessing WEOM sources and humification and will incorporate them in future work by adjusting the analytical protocol.

In the revised Methods section we now added:

The $SUVA_{254}$ and the *E250/E365* ratio are widely used but semi-quantitative indicators. $SUVA_{254}$ can be inflated by non-aromatic UV-absorbers such as nitrate and dissolved Fe(III) and cannot distinguish among different aromatic moieties (Weishaar et al., 2003; Logozzo et al., 2022). The *E250/E365* ratio provides only a coarse estimate of mean chromophore size; it is sensitive to baseline drift, light scattering, and becomes unreliable at low absorbance (Peuravuori and Pihlaja, 2004).

Data Interpretation:

The discussion of negative correlations between BWSOC and environmental factors like $NH_4^+$, $NO_3^-$, and EC is somewhat speculative. The proposed mechanism (e.g., nutrient suppression of microbial degradation) should be more cautiously framed unless further supported.

Response: We agree with the reviewer that the discussion of potential nutrient inhibition of microbial degradation should be presented more cautiously. Accordingly, we have revised the relevant paragraph to acknowledge the speculative nature of this mechanism and to offer alternative explanations. The revised text is as follows:

BWSOC in this study showed a negative correlation with physicochemical parameters. Similar negative EC–BWSOC patterns have been reported for coastal wetland soils, where rising salinity restrained microbial residue accumulation and SOC turnover (Qu et al., 2018; Shao et al., 2022b), and laboratory studies suggest that osmotic stress can curb microbial respiration of labile dissolved organic carbon (Yang et al., 2018). Nutrient effects are strongly context dependent; for instance, nitrogen enrichment reduced soil-carbon mineralization in incubation

experiments with cropland and grassland soils (Perveen et al., 2019), whereas field fertilization in a boreal forest increased DOC concentrations under nitrate addition (Shi et al., 2019). These contrasting findings indicate that multiple, site-specific processes including osmotic stress, stoichiometric imbalance, shifts in microbial community composition, or sorption dynamics may underlie the correlations observed in this study. Further manipulation experiments are required to disentangle these mechanisms.

Clarify whether BWSOC and degradation constants (k) are significantly correlated with SUVA254/E250:E365 using regression statistics.

Response: We thank the reviewer for highlighting the importance of presenting regression statistics. The regression results examining whether BWSOC and the degradation constant ($k$) are significantly correlated with $SUVA_{254}$ and $E250/E365$ are provided below.

[Figure]

Fig S1. Simple linear regressions between the optical indices and the response variables requested by the reviewer. (a) BWSOC vs $SUVA_{254}$; (b) BWSOC vs $E250/E365$; (c) $k$ vs $SUVA_{254}$; (d) $k$ vs $E250/E365$. Grey bands are 95 % confidence intervals. Equations and coefficients of determination ($R^2$) are printed inside each panel.

**Table S1. Simple linear-regression statistics for BWSOC and the degradation constant ($k$) versus the $SUVA_{254}$ and $E250/E365$**

| Outcome | Predictor | Slope $\beta$ | SE | 95 % CI | t | p | $R^2$ |
|---------|-----------|---------------|-----|---------|---|---|-------|
| BWSOC | $SUVA_{254}$ | -0.158 | 0.062 | -0.295 – -0.020 | -2.55 | 0.029 | 0.395 |
| BWSOC | $E250/E365$ | 0.003 | 0.001 | 0.0003 – 0.006 | 2.44 | 0.035 | 0.374 |
| $k$ | $SUVA_{254}$ | 0.014 | 0.262 | -0.570 – 0.599 | 0.06 | 0.957 | 0.000 |
| $k$ | $E250/E365$ | -0.006 | 0.005 | -0.018 – 0.005 | -1.25 | 0.240 | 0.135 |

[Figure]

Fig S2. Observed-versus-predicted plots for the multiple-linear-regression models. (a) BWSOC; (b) degradation constant ($k$).

**Table S2. Model 1 – multiple linear regression predicting BWSOC (g kg$^{-1}$)**

| Predictor | B | SE | $\beta$ | 95 % CI | t | p | VIF |
|-----------|-----|-----|---------|---------|---|---|-----|
| Intercept | 0.203 | 0.059 | - | 0.070 – 0.336 | 3.446 | 0.007 | - |
| $E250/E365$ | 0.002 | 0.002 | 0.354 | -0.002 – 0.006 | 1.115 | 0.294 | 1.705 |
| $SUAV_{254}$ | -0.100 | 0.080 | -0.401 | -0.281 – 0.080 | -1.262 | 0.239 | 1.705 |
| Model fit: n = 12; R = 0.684; $R^2$ = 0.468; Adjusted $R^2$ = 0.350; F = 3.963 | | | | | | | |

**Table S3. Model 2 – multiple linear regression predicting $k$**

| Predictor | B | SE | $\beta$ | 95 % CI | t | p | VIF |
|-----------|-----|-----|---------|---------|---|---|-----|
| Intercept | 0.565 | 0.236 | - | 0.031 – 1.098 | 2.394 | 0.040 | - |
| $E250/E365$ | -0.011 | 0.007 | -0.607 | -0.260 – 0.005 | -1.577 | 0.149 | 1.705 |
| $SUAV_{254}$ | -0.309 | 0.319 | -0.373 | -1.032 – 0.413 | -0.969 | 0.358 | 1.705 |
| Model fit: n = 12; R = 0.466; $R^2$ = 0.217; Adjusted $R^2$ = 0.043; F = 1.245 | | | | | | | |

Simple-linear regressions showed that BWSOC declined with $SUAV_{254}$ ($\beta$ = -0.158 ± 0.062, p = 0.029, $R^2$ = 0.395) and increased with $E250/E365$ ($\beta$ = 0.003 ± 0.001, p = 0.035, $R^2$ = 0.374),

whereas the degradation constant $k$ was not significantly related to either index ($p \geq 0.05$). A multiple model including both optical variables accounted for 35 % of the variance in BWSOC (adjusted $R^2 = 0.350$, $p = 0.058$) but remained non-significant for k (adjusted $R^2 = 0.043$).

Conclusions:

The final statements about climate-driven SOC loss extrapolate from WSOC data. It would be more appropriate to caution that WSOC is a proxy for labile SOC, but not equivalent to total SOC vulnerability under field conditions.

Response: We appreciate the reviewer's insight. We will revise the Conclusions to clarify that WSOC represents the labile portion of SOC and serves as a proxy for microbially accessible carbon, yet it does not fully capture the vulnerability of the entire SOC stock under field conditions. The revised wording more accurately conveys the scope and limitations of our findings.

Revised Conclusions:

This study quantitatively analyzed the biodegradability of water-soluble organic carbon (WSOC) at various depths in a boreal forest. Our results show that BWSOC content ranges from 0.089 g/kg to 0.321 g/kg, with the lowest observed biodegradable WSOC in surface soil WSOC reaching 68.08%. Three-dimensional fluorescence spectroscopy indicated that surface WEOM is dominated by highly aromatic, humic-acid-like matter. The proportion of fulvic-acid-like compounds increased with depth, whereas WSOC aromaticity and molecular weight declined. As a result, biodegradable SOC in soils below 2 m reached 80.8%. Although WSOC in deep horizons had a lower degradation rate than in the upper profile, it remained highly biodegradable. Correlation analyses indicate that the molecular composition of WSOC is a key factor influencing its biodegradability. Overall, WSOC contents at the southern margin of the boreal forest were comparable to those in higher latitudes. Because WSOC represents the most labile fraction of the SOC pool, our results suggest that continued climate warming may

accelerate losses of labile SOC throughout the soil profile in boreal forests.

Technical Corrections

Abstract: Define "BWSOC" at first use (L16).

Response: Biodegradable WSOC (BWSOC) denotes the portion of water-soluble organic carbon that can be utilized and metabolized by microorganisms (Khan et al., 1998; Marschner and Kalbitz, 2003; Scaglia and Adani, 2009; Vonk et al., 2015).

L60–61: "in situ conditions" – Specify if this refers to field conditions vs. extracted WSOC.

Response: We appreciate this clarification request. In the revised manuscript we will make it explicit that, in our study, "in situ" refers to laboratory analyses of WSOC solutions extracted from the corresponding soil depths, rather than direct field measurements.

L194: "formulas ... in Supporting Information" – Ensure these are provided.

Response: We have verified that all formulas are already provided in the Supporting Information.

Table 2: Units: "g/kg" should be "g kg$^{-1}$" for consistency; "%" should be "%.

Response: Thank you for pointing this out. In accordance with the journal's format, we have changed "g/kg" to "g kg$^{-1}$" and ensured that "%" is consistently presented as "%.

Fig. 2: Axis labels are garbled (e.g., "8-10cm" vs. "0-10cm"). Correct depth labels and ensure variables are clearly defined in the caption.

Response: We will correct the depth labels in Fig. 2 and ensure that all variables are clearly

defined in the caption.

Fig. 6: Y-axis label "BWSOC (g/kg)" → "BWSOC (g kg⁻¹)".

Response: The Y-axis label in Fig. 6 has been changed from "BWSOC (g/kg)" to "BWSOC (g kg⁻¹)"。

L263–264: Incomplete sentence ("resulted in very...").

Response:   The full, corrected sentence now reads: The WSOC content at depths of 60-120 cm and the absorbance values at depths of 220-740 cm were extremely low by day 28, which likely resulted in very low SUVA$_{254}$ and *E250/E365* values on that day. As a result, we excluded these data from the analysis.

Referee 2

This manuscript provides a useful exploration of water soluble organic carbon from boreal forest soils; the extreme depths of the cores used for the incubation experiments are quite novel and offer a significant contribution to research on soil biochemistry, hydrology, and DOM dynamics overall. However, there are some potential methodological issues that need clarification to properly contextualize the findings. The difference in time between creating the WSOC mixtures for the incubations (4 hours) and the fluorescence intensity analysis (24 hours) creates a margin of error that requires constraining the interpretation of these results more strongly. Additionally, it is difficult to tell if the nutrient addition concentrations are appropriate for each layer given the data available in the manuscript. I believe this manuscript exhibits excellent scientific significance and good scientific and presentation quality. I recommend publication with minor revisions.

Response: We sincerely thank Reviewer 2 for their encouraging evaluation and insightful suggestions. We appreciate the recognition of our deep-profile design and the potential

contribution to the understanding of WSOC dynamics in boreal soils. In response, we have revised the manuscript to address all raised concerns regarding methodology, terminology, and data contextualization. Below we provide point-by-point responses.

Line 71: typo? Believe "object" should be "objective"

Response: Thank you to the reviewer for noting this typographical error; we have corrected "object" to "objective."

Line 101: citation for the gravimetric soil moisture method if possible.

Response: we will add the relevant reference for the gravimetric soil-moisture determination method (Reynolds, 1970) to the manuscript.

Line 104-106: Was the soil oven dried before combustion?

Response: Yes. We have clarified this in the revised manuscript as follows: the TC and SOC samples were first air-dried. Before SOC determination, approximately 100 mg of sample was weighed into a ceramic boat, an excess of 4 mol $L^{-1}$ HCl was added until no bubbles evolved, the mixture was thoroughly homogenized and left to stand for 4 hours, and then dried at 65°C for 16 hours prior to analysis (Nelson and Sommers, 1996).

Lines 107-110, 139-144: Could you clarify the difference between WSOC and WEOM (incubation time of extractions) for readers? Why were EEMs the only analysis conducted on the WEOM? Are there any data available from the WEOM extractions on C/N or anything else? It's quite possible the longer incubation time for these extractions would result in a different composition of leachate from the four hour incubated samples, and so the EEMs may not be wholly comparable to parameters collected in the WSOC samples. The justification for this is important for contextualizing results like those of lines 220-222. It is possible the fluorescence intensity of these areas may not translate to the availability of these classes of compounds in

the experimental incubation solutions. The degradability of WSOC in layers 60-180 shown in table 2 is well supported by the results shown in figure 3. I suspect this is largely due to the results shown by the EEMs analysis (figure 5), which appear to show larger ratios of fulvic acid to humic acid, but this again needs to be further contextualized given the differences in preparation between the incubations and the samples used for the EEMs. I find these data to correlate well and be quite convincing despite the preparation discrepancy, so my suggestion is to provide justification for the difference and briefly discuss the possibility for error in the conclusions rather than to exclude data generated from the EEMs.

Response: We thank the reviewer for these insightful and constructive comments. Below, we clarify in the Methods section why the extraction times for WSOC and WEOM differ. In addition, because our primary goal was to reveal how depth-dependent differences in WEOM chemical composition influence biodegradability, we believe the WEOM data remain highly valuable. We therefore retain the EEM results; however, we acknowledge that future work should supplement them with additional analyses.

Added to the Methods section:

Because solute concentrations are lower in deeper-soil extracts, the extraction time for WEOM was extended relative to that for WSOC to obtain sufficient concentration and fluorescence signal (Zhou et al., 2023). Although a longer extraction can introduce minor compositional changes (Corvasce et al., 2006; Park and Snyder, 2018), EEM fluorescence nevertheless remains a robust method for assessing WSOC biodegradability (Vonk et al., 2015; Mu et al., 2017; Zhou et al., 2023).

Revisions to the Discussion:

Water-soluble organic carbon (WSOC) in boreal forest soils is highly biodegradable, with the largest proportion of biodegradable WSOC consistently occurring at depths of 60-160 cm. This depth-dependent pattern was reproduced in all replicates. In this study, the 60-160 cm interval

has a buried organic horizon that contains high organic-matter concentrations (Werdin-Pfisterer et al., 2012). Spectroscopic results further confirm that the depth-related differences in BWSOC % arise from variations in the chemical composition of water-extractable organic matter. In Alaskan Kolyma River basin, WSOC concentrations decreased by about 50% following a seven-day incubation (Spencer et al., 2015); in deep soils, ancient low-molecular-weight organic acids within WSOC are rapidly mineralized, leading to a ~53% decline in WSOC after 200 hours of incubation (Koven et al., 2015). The high biodegradability of WSOC is closely related to its chemical composition (Burd et al., 2020). In these regions, WSOC primarily consists of low-aromaticity, low-molecular-weight organic matter that is readily decomposed by microbes (Drake et al., 2015), making it easily accessible for microbial utilization (Ward and Cory, 2015).

Revisions to the Conclusions:

This study quantitatively analyzed the biodegradability of water-soluble organic carbon (WSOC) at various depths in a boreal forest. Our results show that BWSOC content ranges from 0.089 g/kg to 0.321 g/kg, with the lowest observed biodegradable WSOC in surface soil WSOC reaching 68.08%. Three-dimensional fluorescence spectroscopy indicated that surface WEOM is dominated by highly aromatic, humic-acid-like matter. The proportion of fulvic-acid-like compounds increased with depth, whereas WSOC aromaticity and molecular weight declined. As a result, biodegradable SOC in soils below 2 m reached 80.8%. Although WSOC in deep horizons had a lower degradation rate than in the upper profile, it remained highly biodegradable. Correlation analyses indicate that the molecular composition of WSOC is a key factor influencing its biodegradability. Overall, WSOC contents at the southern margin of the boreal forest were comparable to those in higher latitudes. Because WSOC represents the most labile fraction of the SOC pool, our results suggest that continued climate warming may accelerate losses of labile SOC throughout the soil profile in boreal forests.

Line 126, 225, 257, 274, 348 and figure 9: typo in SUVA254. Please check throughout.

Response: All spelling errors have been corrected.

The introduction section does not adequately connect to the methods section. What is the purpose of preparing microbial inocula (lines 159-168)? Was the inocula from a given soil layer simply returned to the same soil layer's WSOC, or was there a cross-inoculated treatment? Could you please provide a stronger transition between the introduction and the purpose of these inocula (i.e.; the role of microbes in WSOC turnover and section 3.3 of the results). Given the interesting results presented in figure 9 between soil moisture and most of the measured variables, I think a background discussion on factors influencing microbial metabolism is warranted.

Response: We thank the reviewer for pointing out the weak linkage between the Introduction and Methods sections, especially regarding the purpose and rationale for preparing the microbial inoculum. In the final paragraph of the Introduction we now emphasize the pivotal role of microbes in WSOC turnover and add the requested background discussion on factors that influence microbial metabolism. The revised text reads:

Microorganisms play a key role in the carbon cycle and strongly influence the biodegradability of WSOC (Marschner and Kalbitz, 2003; Kalbitz et al., 2003; Neff and Asner, 2001; Yano et al., 2000). Microbial biomass is more abundant in surface horizons, where soil-organic-carbon mineralization proceeds rapidly (Henneron et al., 2022; Pei et al., 2025), whereas thaw-activated bacteria in deeper layers can rapidly mineralize WSOC after permafrost thaws (Drake et al., 2015). Microbial use of WSOC is modulated by environmental factors such as soil moisture (Zhang et al., 2024; Li et al., 2020) and soil physicochemical properties (Lv et al., 2024; Shao et al., 2022a). Therefore, detailed knowledge of the content, chemical composition, and biodegradability of WSOC along deep soil profiles is critical for clarifying how subsurface carbon is mobilized, transformed, and ultimately influences carbon cycling in boreal forests. The objective of this study is to quantify WSOC of soil profile that deeper than 3 m in a southern boreal margin. We conducted laboratory incubation experiments to determine differences in

biodegradable water-soluble organic carbon (BWSOC) and employed spectroscopic techniques to reveal its compositional characteristics (Kothawala et al., 2014; Chavez-Vergara et al., 2014; Sun et al., 2022; Murphy et al., 2008; He et al., 2023). The results can improve our understandings of SOC in boreal forests under a warming climate.

In the Methods section we now clarify that the microbial inoculum was prepared by gently extracting the microbial community from each soil layer and adding it to the WSOC solution extracted from the same layer (i.e., no cross-layer inoculation). The revised sentence is:

Inocula and WSOC samples were always matched by depth, preserving the natural microbe–substrate association that is critical for realistic assessments of WSOC bioavailability and degradation potential (Bhattacharyya et al., 2022; Pei et al., 2025).

Lines 179-180: This method doesn't necessarily standardize nutrient availability. The starting concentration of C:N:P in each layer could result in oversaturation in a given incubation; especially given the fairly high nutrient concentrations in the top 10 cm (Figure 1). In fact, at least for C:N ratios, it looks like the top 60 cm are all above redfield ratio. This makes sense given the manuscript's findings in lines 216-219. Given that the concentrations of nutrient solutions in lines 169 and onwards are in molarity and the concentrations of C and N in the starting soils (Figure 1) are in mg/kg, it's not easy to confirm if these amendment ratios are appropriate for the ambient C:N:P of each layer's WSOC incubations. Given that these depths also exhibited relatively lower BWSOC % than other layers (figure 6), it raises the question of possible saturation. However, the optical data provide support for the conclusion that the findings are related to the soil OC and not a methodological issue. Thus, I suggest either demonstrating the amendments were in appropriate concentrations or further constraining the conclusions section to account for a margin of error.

Response: Thank you for your suggestion. In response, we have further constrained the Conclusions section to acknowledge this margin of error.

Additions to the Methods section:

Previous studies suggested that these additions are sufficient to prevent nutrient limitation and to standardize microbial activity (Mehring et al., 2013; Helton et al., 2015). By equalizing nutrient supply across soil layers, we attribute any differences in WSOC consumption to the intrinsic properties of the WSOC itself. We caution, however, that the nutrient amendment may overestimate BWSOC relative to natural field conditions.

Additions to the Discussion section:

Uniform nutrient supply could have induced nutrient-saturation effects in the 0–60 cm samples, where the C/N ratio is likely close to the Redfield ratio, thereby lowering BWSOC % (Aber, 1992; Aber et al., 1997; Gress et al., 2007). Moreover, we used depth-matched inocula to preserve native microbe – substrate interactions, consistent with previous WSOC-biodegradability studies (Bhattacharyya et al., 2022; Pei et al., 2025; Vonk et al., 2015), while we did not quantify microbial abundance or community composition. Therefore, the high BWSOC % values reflect potential rather than in-situ decomposition rates. Future work is required to examine in-situ nutrient status, microbial mass and microbial community structure to better understanding the depth-dependent WSOC dynamics.

Revised, more cautious Conclusions:

This study quantitatively analyzed the biodegradability of water-soluble organic carbon (WSOC) at various depths in a boreal forest. Our results show that BWSOC content ranges from 0.089 g/kg to 0.321 g/kg, with the lowest observed biodegradable WSOC in surface soil WSOC reaching 68.08%. Three-dimensional fluorescence spectroscopy indicated that surface WEOM is dominated by highly aromatic, humic-acid-like matter. The proportion of fulvic-acid-like compounds increased with depth, whereas WSOC aromaticity and molecular weight declined. As a result, biodegradable SOC in soils below 2 m reached 80.8%. Although WSOC in deep horizons had a lower degradation rate than in the upper profile, it remained highly

biodegradable. Correlation analyses indicate that the molecular composition of WSOC is a key factor influencing its biodegradability. Overall, WSOC contents at the southern margin of the boreal forest were comparable to those in higher latitudes. Because WSOC represents the most labile fraction of the SOC pool, our results suggest that continued climate warming may accelerate losses of labile SOC throughout the soil profile in boreal forests.

Is there data available for the starting C, N, and P concentrations at Day 0, either in molarity or in mg/L? A table of these values might be an appropriate addition. A brief discussion of the data as potential rather than actual rates (lines 326-333) is presented but could be bolstered by a discussion of the above issue.

Response: We thank the reviewer for raising this important point. We acknowledge that we did not directly measure the starting concentrations of C, N, and P in the WSOC incubation solutions and therefore cannot provide Day-0 values in mol $L^{-1}$ or mg $L^{-1}$. This limitation stems from our experimental design, which was intended to assess potential biodegradability under standardized, nutrient-replete conditions rather than to replicate ambient field stoichiometry.

We have added an explicit note in the Discussion to draw attention to this limitation for future work, as follows:

Therefore, the high BWSOC % values reflect potential rather than in-situ decomposition rates. Future work is required to examine in-situ nutrient status, microbial mass and microbial community structure to better understanding the depth-dependent WSOC dynamics.

In section 3.3., could you please clarify terms like "deeper soils" and "upper layers." "Deep soils" can refer to layers as shallow as 50 cm depending on the experimental application. Given the impressive core depths used in these incubations, defining terms like "deep" somewhere in the methods section could help clarify these results. Otherwise, perhaps change terms like "upper layers" to specific depth increments like 0-60 cm, where appropriate.

Response: The revised text for section 3.3 is as follows:

BWSOC content and degradation kinetics exhibited pronounced, non-linear depth patterns (Fig. 6). Soils at 60-180 cm depth exhibited higher BWSOC content and degradability compared to other depths (Fig. 6). Significant variations in degradation rates were observed during the incubation process. The reaction kinetics constant ($k$ values) indicated that WSOC degradation in deeper soils (220-740 cm) proceeded more slowly (0.0681-0.0863 $day^{-1}$) (Table 2), occurring predominantly between days 14 and 28 of incubation. In contrast, the WSOC at 60-90 cm depth decomposed rapidly during the early stages of incubation, with a $k$ value of 1.0952 ($day^{-1}$). In summary, although deeper soils (below 2m) also contain relatively high BWSOC content, decomposition in these layers occurs primarily during the later stages of incubation (days 14–28), whereas the WSOC in upper layers (0-180 cm) was rapidly decomposed at the beginning of the incubation period (Fig. 7).

Line 277: I suggest changing "other environmental factors" to "other physicochemical parameters" or similar for clarity. "Environmental factors" traditionally invokes things like precipitation, land cover, etc., rather than soil nutrient concentrations or EC. Possibly suggest a similar change in lines 344-351.

Response:    Thank you for pointing this out; we have made the correction.

[revised manuscript text omitted]